CIT-EmotionNet: convolution interactive transformer network for EEG emotion recognition

Lu Wei 1 2 3
Xia Lingnan 1
Tan Tien Ping 2 tienping@usm.my
Ma Hua 1 3 mahua11352@outlook.com
1 Henan High-Speed Railway Operation and Maintenance Engineering Research Center, Zhengzhou Railway Vocational and Technical College , Zhengzhou, Henan , China
2 School of Computer Sciences, Universiti Sains Malaysia , USM, Pulau Pinang , Malaysia
3 Zhengzhou University Industrial Technology Research Institute , Zhengzhou, Henan , China
Murugappan M.
Electronic publication date: 2024 Dec 23
Publication date: 2024
Volume: 10
Electronic Location ID: e2610
Received 2024 Feb 13; Accepted 2024 Nov 25
Copyright: © 2024 Lu et al.
Copyright year: 2024
Copyright holder: Lu et al.
License: This is an open access article distributed under the terms of the Creative Commons Attribution License, which permits unrestricted use, distribution, reproduction and adaptation in any medium and for any purpose provided that it is properly attributed. For attribution, the original author(s), title, publication source (PeerJ Computer Science) and either DOI or URL of the article must be cited.
License URL: https://creativecommons.org/licenses/by/4.0/

Keywords: Affective computing, Electroencephalogram (EEG), Emotion recognition, Convolutional neural network (CNN), Transformer

Funding: Henan Provincial Science and Technology Research Project, China 232102240091, 232102240089, and 242102241064 Key Scientific Research Project of Henan Province Higher Education Institutions, China 23B520033 and 25B580004 Research Project of Zhengzhou Railway Vocational and Technical College, China 2022KY008 and 2022KY015 This research was funded by the Henan Provincial Science and Technology Research Project, China (Grant Nos. 232102240091, 232102240089, and 242102241064), Key Scientific Research Project of Henan Province Higher Education Institutions, China (Grant Nos. 23B520033 and 25B580004), and the Research Project of Zhengzhou Railway Vocational and Technical College, China (Grant Nos. 2022KY008 and 2022KY015). The funders had no role in study design, data collection and analysis, decision to publish, or preparation of the manuscript.

==============================
Emotion recognition is a significant research problem in affective computing as it has a lot of potential areas of application. One of the approaches in emotion recognition uses electroencephalogram (EEG) signals to identify the emotion of a person. However, effectively using the global and local features of EEG signals to improve the performance of emotion recognition is still a challenge. In this study, we propose a novel Convolution Interactive Transformer Network for EEG Emotion Recognition, known as CIT-EmotionNet, which efficiently integrates the global and local features of EEG signals. We convert the raw EEG signals into spatial-spectral representations, which serve as the inputs into the model. The model integrates convolutional neural network (CNN) and Transformer within a single framework in a parallel manner. We propose a Convolution Interactive Transformer module, which facilitates the interaction and fusion of local and global features extracted by CNN and Transformer respectively, thereby improving the average accuracy of emotion recognition. The proposed CIT-EmotionNet outperforms state-of-the-art methods, achieving an average recognition accuracy of 98.57% and 92.09% on two publicly available datasets, SEED and SEED-IV, respectively.

Introduction

Emotion recognition (Jia et al., 2021; Tan et al., 2020; Cimtay, Ekmekcioglu & Caglar-Ozhan, 2020; Doma & Pirouz, 2020) has become a significant research task in the field of affective computing due to its potential applications in various areas, including affective brain-computer interfaces, the diagnosis of affective disorders, emotion detection in patients with consciousness disorders, emotion detection of drivers, mental workload estimation, and cognitive neuroscience. Emotion is a mental and physiological state that arises from various sensory and cognitive inputs, significantly influencing human behavior in daily life (Jia et al., 2021). Emotion is a response to both internal and external stimuli (Cheng et al., 2024). Physiological signals, such as electrocardiogram (ECG), electromyography (EMG), and electroencephalogram (EEG), correspond to the physiological responses caused by emotions. They are more reliable indicators of emotional expression than non-physiological signals, such as speech, posture, and facial expression, which can be masked by humans (Tan et al., 2020; Cimtay, Ekmekcioglu & Caglar-Ozhan, 2020). Among these physiological signals, EEG signals have a high temporal resolution and a wealth of information, which can reveal subtle changes in emotions, making them more suitable for emotion recognition than other physiological signals (Atkinson & Campos, 2016). EEG-based emotion recognition methods are more accurate and objective, as some studies have verified the relationship between EEG signals and emotions (Xing et al., 2019).

The convolutional neural network (CNN) (Kwon, Shin & Kim, 2018; Shen et al., 2023a; Salama et al., 2018; Shen et al., 2021) is well-known for its efficient feature extraction and powerful feature representation ability. Many researchers use CNN for emotion recognition from electroencephalography (EEG) signals. For example, Kwon, Shin & Kim (2018) introduced a CNN model for EEG feature extraction. Yang, Han & Min (2019) proposed a multi-column CNN-based model for emotion recognition from EEG signals. Chen et al. (2019) presented a CNN-based method for learning and classifying EEG emotion features. However, EEG signals are sequential data. The local receptive fields of CNN-based methods may struggle to accurately extract the global features of EEG signals that contain sequential information (Li et al., 2022b).

On the other hand, Transformer has also been used in emotion. For example, Liu et al. (2022) proposed four variant transformer frameworks for EEG emotion recognition to investigate the relationship between emotion and spatial-temporal EEG features, making the traditional Transformer model suitable for EEG signals. Wang et al. (2023) introduced the Joint-Dimension-Aware Transformer (JDAT), a robust model for EEG emotion recognition that utilizes a compressed Multi-head Self-Attention mechanism. However, unlike convolutional operations used in CNN models, the Transformer model utilizes a self-attention mechanism to model the dependencies between elements in the sequence. While the self-attention mechanism is capable of capturing global contextual information, it may not be as effective as convolutional operations in extracting local features that include spatial information (Zhao et al., 2023).

To improve the accuracy of the model, researchers are exploring effective strategies to leverage the advantages of both CNN and Transformer models. In the field of visual recognition, Peng et al. (2021) introduced a network structure called Conformer, which combines both CNN and Transformer models, to leverage the benefits of convolutional operations and self-attention mechanisms for improved representation learning. In the field of EEG-based recognition, Zhao et al. (2023) introduced an approach for seizure detection that incorporates interactive local and global features. The model utilizes convolutional operations and self-attention mechanisms to extract both local features and global representations from EEG signals. Zhou et al. (2024) introduced an EEG model, GlepNet, designed for the automatic detection of epilepsy. By integrating temporal convolution and multi-head attention mechanisms within its encoder, GlepNet effectively captures both local and global features associated with epileptic seizures. Cheng et al. (2024) proposed a Transformer-based model, R2G-STLT, which is capable of learning representative local and global spatiotemporal features from the electrode level to the brain region level. However, the aforementioned model overlooks the fusion of global and local features at each stage, thereby failing to extract more comprehensive feature information. Consequently, effectively integrating the global and local features of EEG signals at different stages to enhance performance in emotion recognition remains a challenging task in the field of EEG emotion recognition.

In order to address the above challenging task, we propose a Convolution Interactive Transformer network for EEG emotion recognition named CIT-EmotionNet. First, we divided the original EEG signal into several segments of 4 s each. For each segment, we extracted the PSD features of the δ, θ, α, β, and γ frequency bands, and mapped the features in space according to the location of the electrodes to generate the EEG feature representation. To extract both local and global features from the EEG representation, we processed the EEG features through both the CNN and Transformer branches. At each stage of the CNN and Transformer branches, we extracted local and global features, respectively. Subsequently, to better fuse and interact with these features, we proposed a Convolution Interactive Transformer module, named the CIT module. The classifier is composed of fully connected layers and softmax layers, which are used to predict the emotional label. Finally, we conducted a series of comparative experiments and ablation experiments using CIT-EmotionNet, which not only proved that CIT-EmotionNet outperformed all state-of-the-art models, but also studied the contribution of key components in CIT-EmotionNet to recognition performance.

The main contributions of this article can be summarized as follows: To enhance the extraction of both local and global features from EEG signals, we have proposed a parallel dual-branch neural network capable of simultaneously capturing these features and integrating them within a unified model.

To better fuse and interact with local and global features, we have developed a CIT module that enables the interaction and fusion of both types of features. This feature interaction and fusion captures finer emotional nuances, enhancing the accuracy of the model.

The proposed CIT-EmotionNet model achieves an average recognition accuracy of 98.57% and 92.09% on the SEED and SEED-IV datasets, respectively, outperforming all state-of-the-art models. In addition, we conducted a series of ablation experiments to investigate the contributions of the key components of CIT-EmotionNet to the recognition performance.

Related work

In this section, we discuss related works in EEG classification, particularly those that apply CNN or Transformer in their model.

CNN-based methods

In recent years, convolutional neural network (CNN)-based approaches (Kwon, Shin & Kim, 2018; Chen et al., 2019; Shen et al., 2019; Li et al., 2022a; Salama et al., 2018; Shen et al., 2022) are known for their capabilities of efficient feature extraction and powerful feature representation. Hence, many researchers have explored using CNN for extracting features from EEG signals for the emotional recognition of electroencephalogram (EEG) signals. Kwon, Shin & Kim (2018) used CNN for feature extraction from EEG signals. In this model, the EEG signal is preprocessed before convolution by wavelet transform while both the time and frequency of the EEG signal are taken into account (Kwon, Shin & Kim, 2018). Yang, Han & Min (2019) presented a multi-column CNN-based model for emotion recognition from EEG signals. The multi-column CNN-based model, whose decision was produced by a weighted sum of the decisions from individual recognizing modules (Yang, Han & Min, 2019). Based on the time and frequency combination EEG features, Chen et al. (2019) proposed a deep CNN model similar to that used for image classification in computer vision. The model not only avoided the labor-intensive process of manual feature extraction and feature selection prior to traditional machine learning classification but also effectively improved the accuracy and stability of EEG emotion recognition (Chen et al., 2019). Liu et al. (2020) came up with a kind of dynamic differential entropy (DDE) algorithm to extract the features of EEG signals. Subsequently, the extracted DDE features were classified by using a CNN (Liu et al., 2020). Rahman et al. (2021) suggested transforming EEG signals into brain topographic mapping covering frequency and spatial information, and classifying emotions by using CNN. Gilakjani & Al Osman (2022) proposed a deep learning model that combines CNN and residual recurrent neural networks (RNN) to extract both spatial and temporal features from multi-channel EEG signals. Not only 2D CNN have achieved good results in terms of EEG emotion recognition tasks, but 3D CNN has done remarkable work. For instance, Salama et al. (2018) investigated the application of 3D convolutional neural networks (3D-CNN) in emotion recognition and developed a data enhancement stage to improve the performance of 3D CNN models. Cho & Hwang (2020) presented a 3D CNN model that can efficiently represent the spatial-temporal representation of EEG signals for emotion recognition. Although CNN has achieved remarkable success in recognizing emotions from EEG signals and possesses advantages in local feature extraction, it is a locally sensitive model where each convolutional kernel only focuses on local features. This tends to result in the neglect of global features during feature extraction, making it difficult to capture the sequential information in EEG signals. This limitation can affect its performance in EEG emotion recognition tasks (Zhao et al., 2023).

Transformer for EEG classification

Transformer based on global self-attention is gaining increasing attention from researchers. The Transformer was first proposed to apply in the field of natural language processing, then it was introduced to the field of computer vision (Han et al., 2023). Vision Transformer (ViT), a complete self-attention-based image recognition model, has obtained state-of-the-art results in object detection (Hong et al., 2022), image classification (Chen, Fan & Panda, 2021), and segmentation (Gu et al., 2022). There were some studies that apply the Transformer for EEG signals classification. The studies showed that the attention mechanism in a Transformer can capture global information of EEG signal sequences and extract more relevant features for classification (Wang et al., 2023). For epileptic seizure prediction based on EEG signals, Hussein, Lee & Ward (2022) introduced a Transformer-based approach named Multi-Channel Vision Transformer (MViT) to automatically and simultaneously learn spatio-temporal-spectral features in multichannel EEG data. For the task of emotion recognition based on EEG signals, Liu et al. (2022) proposed an EEG emotion Transformer (EeT) model that learns spatial-spectral features directly from EEG signal sequences, thus making the traditional Transformer model suitable for EEG signals. Wang et al. (2023) introduced a model called Joint-Dimension-Aware Transformer (JDAT) for EEG emotion recognition, which was based on the squeezed multi-head self-attention (MSA) mechanism. By applying adaptive squeezed MSA to multidimensional features, JDAT was able to focus on diverse EEG information, including space, frequency, and time (Wang et al., 2023). Wang et al. (2022b) proposed a Transformer-based model for hierarchical learning of differentiating spatial information from the electrode level to the brain region level. Although the above-mentioned models have shown the effectiveness of Transformer models in EEG classification tasks, it should be noted that Transformer models are not particularly adept at handling local features (Zhao et al., 2023). Therefore, using only a Transformer model may overlook spatial information in EEG signals.

CNN and transformer fusion

To draw on the advantages of the CNN model featuring local information extraction and the Transformer model characterized by global information extraction, researchers (Zhang, Liu & Hu, 2021; Jiang et al., 2022; Shen et al., 2023b; Jiang et al., 2022; Song, Liu & Ma, 2022) have attempted to fuse these two models. In the field of visual recognition, Peng et al. (2021) introduced a novel network architecture called Conformer, which combined the strengths of convolutional operations and self-attention mechanisms to improve representation learning. Conformer was built upon the feature coupling unit (FCU), which interactively integrated local features and global feature representations at different resolutions (Peng et al., 2021). In the field of medical images, Zhang, Liu & Hu (2021) proposed a parallel TransFuse model to combine Transformer and CNN, in which both global dependency and low-level spatial details can be efficiently captured in a shallower way. In addition, a new fusion technique, BiFusion module, was proposed to efficiently fuse the multilevel features of both branches (Zhang, Liu & Hu, 2021). The TD-Net model proposed by Song, Liu & Ma (2022) combines Transformer and CNN to extract global and local information, using the former as an encoder to extract the global information and the latter to obtain the feature mapping of the image. The coding results from the Transformer are then connected to the CNN upsampling process (Song, Liu & Ma, 2022). Jiang et al. (2022) designed a new network model MTPA_Unet which combines Transformer and CNNs in a serial manner. In the field of remote sensing image semantic segmentation, Gao et al. (2021) proposed the STransFuse model that combines the advantages of both Transformer and CNN, which uses a staged model to extract coarse-grained and fine-grained feature representations at different semantic scales. Wang et al. (2022a) presented a Coupled CNN and Transformer Network (CCTNet) that combines the local details of the CNN and the global context of the Transformer. This model fuses the features of CNN and Transformer branches while maintaining the advantages of each. Additionally, the light adaptive fusion module (LAFM) and the coupled attention fusion module (CAFM) are also proposed to smoothly fuse the features of CNN and Transformer (Wang et al., 2022a). Zhang et al. (2022) put forward a model fusing Transformer and CNN, in which, the encoder module extracts features through Swin Transformer and the decoder module draws on some effective blocks from the CNN model. In the field of EEG-based recognition, Zhao et al. (2023) proposed an interactive approach for seizure detection that combines both local and global features. Local features are extracted using convolution operations, while global representations of the EEG signals are obtained through a self-attention mechanism. However, the previously mentioned model neglects the fusion of global and local features at each stage, resulting in a limited extraction of comprehensive feature information. Therefore, effectively integrating the global and local features of EEG signals at different stages to improve performance in emotion recognition remains a formidable challenge in the domain of EEG emotion recognition.

Methodology

In this section, the proposed network is presented in six parts: preliminaries, model overview, EEG feature representation, CNN branch, Transformer branch, and CIT module. Portions of this text were previously published as part of a preprint (https://doi.org/10.48550/arXiv.2305.05548) (Lu, Ma & Tan, 2023).

Overview

In this article, we utilize time-frequency feature extraction methods to derive the power spectral density (PSD) features across five frequency bands {δ,θ,α,β,γ} from all EEG channels in the EEG signal samples. The formulation of PSD is presented in Eq. (1).

(1) PSD=E[x2],

we define S=(S1,S2,...,SB)∈RNe×B as the frequency features containing B frequency bands extracted from EEG signals, where Ne is the number of electrodes. Based on the electrode positions on the scalp, we construct the spatial-spectral representation SM=(S1M,S2M,...,SBM)∈RH×W×B, where H and W denote the height and width of the frequency map respectively. {δ,θ,α,β,γ} is selected as the frequency band set.

The overview of the proposed CIT-EmotionNet model structure is depicted in Fig. 1. The raw EEG signals are first converted into EEG feature representations. Subsequently, these EEG feature representations are fed into CIT-EmotionNet for EEG emotion recognition. CIT-EmotionNet is primarily composed of three parts: CNN branch, Transformer branch, and CIT module. The EEG feature representations are input to the CNN branch and Transformer branch. The CNN branch consists of five stages, namely stage 0, stage 1, stage 2, stage 3, and stage 4. Each stage of the CNN branch from stage 1 to stage 4 contains two ResNet basic blocks, which respectively output local features C1, C2, C3, and C4. Similarly, the Transformer branch also consists of five stages, namely stage 0, stage 1, stage 2, stage 3, and stage 4. Each stage of the Transformer branch from stage 1 to stage 4 contains three Transformer Encoder blocks, which respectively output global features T1, T2, T3, and T4. The local features C1, C2, and C3, as well as the global features T1, T2, and T3, are input into the CIT module of stages 1, 2, and 3, respectively, where they are interactively fused. The CIT module comprises two blocks: the global to local block and the local to the global block. The output of the CIT module for each stage is the input for the next stage of both the CNN branch and the Transformer branch. Specifically, the output of the global to local block corresponds to the input for the next stage of the CNN branch, while the output of the local to global block corresponds to the input for the next stage of the Transformer branch. Finally, the local features C4 from stage 4 of the CNN branch and the global features T4 from stage 4 of the Transformer branch are merged through a Concat module, which is then fed into a fully connected layer for emotion classification recognition.

Figure 1 Overview structure of the proposed CIT-EmotionNet.

CIT-EmotionNet consists of three main components: the CNN branch, the Transformer branch, and the CIT module. The CNN branch is primarily responsible for extracting local features, the Transformer branch is mainly responsible for extracting global features, and the CIT module is mainly used to achieve the interaction and fusion between global and local features.

EEG feature representation

Figure 2 shows the process of converting the original EEG signals into spatial-spectral representations. Data were collected as previously described in Lu, Tan & Ma (2023). Specifically, we divide the original EEG signals into non-overlapping periods lasting for 4 s, and each segment is assigned the same label as the original EEG signals.

Figure 2 Process of converting original EEG signal into a feature representation.

Firstly, the raw EEG signal is segmented into fixed-length segments. Secondly, frequency-domain features of different frequency bands are extracted from each segment. Finally, these features are mapped to the electrode position matrix to construct the EEG feature representations.

To construct the EEG feature representation, the temporal-frequency feature extraction method is used to extract the PSD features of five frequency bands {δ,θ,α,β,γ} of all EEG channels from the EEG signal samples in the EEG segments with a length of 4 s. We define S=(S1,S2,...,SB)∈RNe×B as a frequency feature containing a frequency band extracted from the PSD feature, in which the frequency band is B∈{δ,θ,α,β,γ}, the electrode is Ne∈{FP1,FPZ,...,CB2}, and XbB=(xb1,xb2,...,xbN)∈RN,(b∈{1,2,...,B}) represent the collection of EEG signals from all Ne electrodes on the frequency band B. Then, the selected data are mapped to a frequency domain brain electrode location matrix SbM∈RH×W,(b∈{1,2,...,B}) according to the electrode location on the brain. Here, the superscript M indicates that this feature is represented as a feature matrix. Finally, the frequency-domain brain electrode position matrices from different frequencies are superimposed to form the spatial-spectral feature representation of EEG signals, that is, the construction of the EEG feature representation SM=(S1M,S2M,...,SBM)∈RH×W×B is finished.

CNN branch

The Residual Network (ResNet) was proposed to address the problem of degradation in deep CNN models. ResNet utilizes residual connections to link different convolutional layers, thereby enabling the propagation of shallow feature information to the deeper layers. Thus, it is suitable to be used for local feature extraction from EEG feature representation. The specific structure of the CNN branch is shown in Fig. 3.

Figure 3 The specific structure of CNN branch.

The CNN branch consists of five stages, where stage 0 preprocesses the input EEG feature representation with operations such as convolution, pooling, and batch normalization. Each of stage 1, stage 2, stage 3, and stage 4 consists of two BasicBlocks, which are the core building blocks of ResNet. Each BasicBlock consists of a convolutional layer, batch normalization (BN) layer, and a non-linear activation function (ReLU).

The input of ResNet is the spatial-spectral feature representation SM=(S1M,S2M,...,SBM)∈RH×W×B. The spatial-spectral feature representation first goes through Stage 0, which consists of a 7×7 convolutional layer, a max pooling operation, and batch normalization. The purpose of this stage is to perform initial convolutional feature extraction and downsampling of the EEG feature representation, reducing the size of the feature map while increasing the level of abstraction. Stage 0 preprocesses the input EEG feature representation through convolution, pooling, and batch normalization operations, transforming it into a smaller and more abstract feature representation, providing better input features for subsequent convolutional blocks. Specifically, the input of Stage 0 is the spatial-spectral feature representation SM=(S1M,S2M,...,SBM)∈RH×W×B, where the shape of the spatial-spectral feature representation is H×W×C, with H representing the height, W representing the width, and C representing the number of channels. Since the number of frequency bands is 5, C=5. However, this does not meet the input requirements of the original ResNet model since the first convolutional layer in the original ResNet model requires three input channels. If we directly use the original model to process data with five input channels, channel conversion or padding operations are required, which can be tedious. Therefore, we replaced the first half of the ResNet model with a new convolutional layer that has five input channels, 64 output channels, a kernel size of 7×7, a stride of 2, a padding of 4, and no bias. After passing through the new convolutional layer, the size and number of the output feature maps have changed. The equation for calculating the output shape after features pass through a convolutional layer is shown in Eq. (2).

(2) output_size=input_size+2∗padding−kernel_sizestride+1.

By substituting the kernel size of 7×7, stride of 2, and padding of 4 into the equation, the shape of the output features after the convolutional layer can be determined. After the convolutional layer, there is another max pooling layer. The calculation formula for the max pooling layer is as shown in Eq. (3).

(3) output_size=input_size−kernel_sizestride+1

By substituting the kernel size of 3×3 and stride of 2 into the equation, the shape of the output features after the max pooling layer can be determined. Due to the need to round down, the division in the formula is more accurately described as finding the quotient. Specifically, after passing through Stage 0, the shape of the output feature map is H4×W4×64. This feature map will be used as the input for the subsequent Stage 1. The equations for Stage 0 of ResNet are shown in Eq. (4).

(4) C0=MaxPool(ReLU(BN(Conv7×7(SM)))),

where SM is the input of Stage 0 in the CNN branch, and C0 is the output of Stage 0 in the CNN branch. Conv7×7(⋅) represents the convolutional layer operation with an output channel of 64, kernel size of 7×7, the stride of 2, and padding of 4. BN(⋅) represents the batch normalization layer operation, which performs batch normalization on the output of the convolutional layer. ReLU(⋅) represents the ReLU activation function, which applies the ReLU activation function to the output of the batch normalization layer. MaxPool(⋅) represents the max pooling layer operation, which performs max pooling using a 3×3 pooling kernel, a stride of 2, and padding of 1.

The features output from Stage 0 are processed through Stage 1, Stage 2, Stage 3, and Stage 4 to generate the C1, C2, C3, and C4 features, respectively. After passing through Stage 1, the shape of the output feature C1 remains the same as the input feature, which is H+82×W+82×64. After passing through Stage 2, the shape of the output feature C2 is H+164×W+164×128. After passing through Stage 3, the shape of the output feature C3 is H+328×W+328×256. After passing through Stage 4, the shape of the output feature C4 is H+6416×W+6416×512. Each of Stage 1, Stage 2, Stage 3, and Stage 4 consists of two BasicBlocks. In BasicBlock, the input feature is added to the main branch output feature via a shortcut connection before being passed through a ReLU activation function. The equation of the main branch, as shown in Eq. (5).

(5) Xmain=BN(Conv3×3(ReLU(BN(Conv3×3(XBasicIN))))),

where XBasicIN is the input of BasicBlock, Xmain is the output of the main branch in the BasicBlock. Conv3×3(⋅) represents the convolutional layer operation with a kernel size of 3×3, the stride of 1, and padding of 1. BN(⋅) represents the batch normalization layer operation, which performs batch normalization on the output of the convolutional layer. ReLU(⋅) represents the ReLU activation function, which applies the ReLU activation function to the output of the batch normalization layer.

The shortcut connection allows the gradient to flow directly through the network, bypassing the convolutional layers in the main branch, which helps to prevent the vanishing gradient problem. The equation of the shortcut connection, as shown in Eq. (6).

(6) Xshortcut==BN(Conv1×1(Conv3×3(XBasicIN))),

where XBasicIN is the input of BasicBlock, Xshortcut is the output of the shortcut connection in the BasicBlock. Conv3×3(⋅) represents the convolutional layer operation with a kernel size of 3×3, the stride of 1, and padding of 1. Conv1×1(⋅) represents the convolutional layer operation with a kernel size of 1×1. BN(⋅) represents the batch normalization layer operation. ReLU(⋅) represents the ReLU activation function.

The addition of the input feature to the main branch output feature allows the network to learn residual mappings, which can be easier to optimize during training. The equation of the addition, as shown in Eq. (7).

(7) XBasicOUT=ReLU(Xmain+Xshortcut),

where Xmain is the output of the main branch, Xshortcut is the output of the shortcut connection, and XBasicOUT is the output of the BasicBlock. ReLU(⋅) represents the ReLU activation function.

Transformer branch

Vision Transformer (ViT) has shown state-of-the-art results in various computer vision applications, such as image classification and segmentation. This motivates us to use ViT to extract global features from EEG feature representation. The structure of the Transformer branch is shown in Fig. 4.

Figure 4 The structure of the Transformer branch.

The Transformer branch consists of five stages, where stage 0 is the input processing stage that divides the input EEG feature representation into several feature representation blocks and obtains initial embedding vectors through linear projection. Stages 1 to 4 are the Transformer Encoder stages, each of which consists of three Encoder blocks used to perform multiple layers of transformation on the embedding vectors to obtain more rich and high-level feature representations.

The input of ViT is the spatial-spectral feature representation SM=(S1M,S2M,...,SBM)∈RH×W×B. The shape of SM is 32×32×5. SM is first processed by Stage 0 and segmented into small blocks, each called a patch. Then, each patch is compressed into a one-dimensional vector, which serves as the input for subsequent self-attention calculations. We set patch_size=32 and hidden_dim=256. Therefore, SM with a shape of 32×32×5 is divided into five patches of size (32×32) each, with a shape of 32×32×1. After Stage 0, each patch is compressed into a vector of size 256, resulting in an output shape of Stage 0 of 5×256.

The features output from Stage 0 are processed through Stage 1, Stage 2, Stage 3, and Stage 4 to generate the T1, T2, T3, and T4 features, respectively. ViT’s Stage 1, Stage 2, Stage 3, and Stage 4 each contain three EncoderBlocks. The EncoderBlock in ViT is designed to establish global context relationships between different positions of the input feature map so that the model can better understand the input feature map. An EncoderBlock comprises of three primary components: multi-head attention, multi-layer perceptron (MLP) block, and layer normalization. Below is a detailed description of each of these components.

Multi-head attention is utilized to perform self-attention operations in different representation subspaces to capture global relationships. Specifically, the Multi-Head Attention first maps the input vector X into three sequences of vectors, namely, Q∈Rn×dq, K∈Rn×dk, and V∈Rn×dv, by three different linear transformations, namely, Query, Key, and Value matrices. Here, dq, dk, and dv are the dimensions of the Query, Key, and Value vectors, respectively. Then, Q, K, and V are transformed by linear mappings of dimension dh to obtain multiple heads H, as shown in Eq. (8).

(8) headi=Attention(QWiQ,KWiK,VWiV),

where, WiQ∈Rdq×dh, WiK∈Rdk×dh, and WiV∈Rdv×dh are parameter matrices of the i-th head, and Attention(⋅) denotes the dot-product attention mechanism.

Dot-product attention is used to compute the similarity between Q and K and weight V by this similarity, as shown in Eq. (9).

(9) Attention(Q,K,V)=Softmax(QKTdk)V,

where, Q, K, and V represent the query, key, and value vectors, respectively, and dk is the dimension of the key vector. The function softmax(⋅) is used to normalize the similarity between the query and key vectors, obtaining the attention weights, which are then used to obtain the weighted sum of the value vectors, resulting in the final output.

To enable the model to learn features in different subspaces and increase its expressive power, the multi-head attention concatenates the output vectors of these attention heads together, and then maps them to a new space through a linear transformation layer, as shown in Eq. (10).

(10) MultiHead(Q,K,V)=Concat(head1,⋯,headh)WO,

where headi represents the output of the i-th attention head, h denotes the number of heads, Concat(⋅) is the operation that concatenates the output of each head, and WO is the linear transformation matrix for the final output.

The MLP Block refers to a multi-layer perceptron module, also known as a fully connected layer module, usually consisting of two linear layers and a GELU activation function. Specifically, the input to each MLP Block is a tensor processed by the multi-head attention module, with a shape of N×L×D, where N represents the number of input sequences, L represents the length of the sequence, and D represents the vector dimension at each position. Then, the MLP Block transforms each vector at each position through two fully connected layers and uses GELU as the activation function. The MLP Block can not only improve the expressive power of the model but also ensure that information is not lost during processing. The MLP Block is shown in Eq. (11).

(11) MLP(x)=Dropout(Dropout(GELU(xW1+b1))W2+b2),

where, x is the input vector, W1 and W2 are the weights of the two fully connected layers, b1 and b2 are the bias terms, GELU(⋅) represents the GELU activation function, and Dropout(⋅) represents the Dropout layer, which is used to reduce overfitting.

Layer normalization is used to normalize inputs to improve the stability and training effectiveness in MLP and Multi-Head Attention modules. Specifically, layer normalization normalizes inputs by subtracting the mean and dividing by the standard deviation of the features. This approach effectively combats the problem of vanishing gradients, without reducing network expressivity, thereby enhancing network stability. The formula of layer normalization is shown in Eq. (12).

(12) LayerNorm(x)=γx−μσ2+ε+β,

where, x represents the input features, γ, and β are learnable scaling and shifting parameters, μ and σ are the mean and standard deviation of the input vector x, and ϵ is a very small value used to avoid the denominator being zero.

CIT module

The local features extracted by the CNN branch can provide more fine-grained information, while the global features extracted by the Transformer branch can provide more comprehensive contextual information. To better integrate local and global features and thus improve the accuracy and robustness of the model, we propose a Convolution Interactive Transformer module, which is called the CIT module. The CIT module enables the interaction and fusion of local and global features, thus enhancing the ability of the model to extract both global and local features from EEG spatial-spectral feature representation, leading to improved performance. The overall framework of the CIT module is shown in Fig. 5.

Figure 5 The overall framework of the CIT module.

The CIT module comprises two main components, namely the L2G block and the G2L block. The L2G block primarily extracts global features from local ones while also integrating the global features derived from the Transformer branch to obtain more enriched global features, which are subsequently utilized as input to the next phase of the Transformer branch. The G2L block mainly converts the global features obtained from the Transformer branch into local features and fuses them with the local features obtained from the CNN branch, resulting in more enriched local features that are then used as input to the next stage of the CNN branch. Through the interaction between the L2G and G2L blocks, the CIT module achieves complementarity and communication between local and global features, thereby enhancing the richness and expressive power of the feature representation.

Every CIT module consists of two blocks: the Local to Global (L2G) block and the Global to Local (G2L) block. The CNN feature maps (Stage k, where k=1,2,3) serve as inputs to the L2G block, which outputs ViT feature maps (Stage k+1, where k=1,2,3). Similarly, the ViT feature maps (Stage k, where k=1,2,3) serve as inputs to the G2L block, which outputs CNN feature maps (Stage k+1, where k=1,2,3). The CIT module can make the local features of the CNN feature map and the global features of the ViT feature map form a coupling state. In other words, global features and local features can interact with each other in the entire feature learning process. The following is a detailed introduction to the L2G block and the G2L block.

In order to improve the ability of the model to extract features from different regions of EEG spatial-spectral feature representation and thereby improve the classification performance, we propose a Convolution Interactive Transformer module, termed the CIT module. The CIT module enables the interaction and fusion of local and global features, thus enhancing the ability of the model to extract both global and local features from EEG spatial-spectral feature representation, leading to improved performance. The overall framework of the CIT module is shown in Fig. 5. Every CIT module consists of two blocks: the Local to Global block, termed L2G, and the Global to Local block, termed G2L. The CNN feature maps (Stage k, where k=1,2,3) serve as inputs to the L2G block, which outputs ViT feature maps (Stage k+1, where k=1,2,3). Similarly, the ViT feature maps (Stage k, where k=1,2,3) serve as inputs to the G2L block, which outputs CNN feature maps (Stage k+1, where k=1,2,3). The CIT module can make the local features of CNN feature map and the global features of ViT feature map form a coupling state. In other words, global features and local features can interact with each other in the entire feature learning process. The following is a detailed introduction to the L2G block and the G2L block.

L2G block

The L2G block is responsible for transforming the input CNN feature maps into ViT feature maps. This block consists of Mean, Linear, Expand, and Concat, and the processing of the L2G block is shown in the following equation.

(13) FGk+1=L2G(FLk),k∈{1,2,3},

where, FGk+1 represents the global feature at stage k+1 and serves as the output to the L2G block, FLk represents the local feature at stage k and serves as the input of the L2G block, and L2G(⋅) represents the transformation process of the L2G block.

L2G(⋅) takes the local features extracted by the CNN branch as input and reduces their dimensionality to match the input dimensionality of the ViT branch. Specifically, it first performs average pooling on the local features to obtain a Batch_size×Channel_num tensor FMean, where Batch_size represents the number of samples in the current batch and Channel_num represents the number of feature dimensions in the current CNN layer. Therefore, FLk in the local to global block first goes through the operation of Mean(⋅), as shown in the following Eq. (14).

(14) FMean=Mean(FLk),

where, the input of the Mean(⋅) operation is FLk, while the output is FMean.

Mean(⋅) performs average pooling on the local feature tensor FLk along the last two dimensions, resulting in a tensor FMeank with a shape of 32×64, where 32 represents the batch size and 64 represents the dimensionality of the local features extracted by the CNN. The Mean(⋅) operation is shown in the following Eq. (15).

(15) Mean(FLk)=1h×w∑m=1h∑n=1wFLm,nk,k∈{1,2,3},

where h represents the height of the feature map, and w represents the width of the feature map.

Then, to facilitate fusion with global features for better performance, L2G(⋅) needs to map the tensor FMean to a space with the same dimensionality as the global features. Specifically, L2G(⋅) applies a linear mapping to FMean using the Linear(⋅) operation, resulting in a tensor FLineark with a shape of 32×768, which has the same dimensionality as the global features. The Linear(⋅) operation is shown in the following Eq. (16).

(16) FLinear=WTFMean+b,

where, the output is FLinear, WT is the linear transformation matrix, and b is the deviation value.

Finally, to preserve the original global features, L2G(⋅) fuses the global feature FGk of stage k with FLinear to obtain the input of the Transformer branch for stage k+1, namely the global feature FGk+1 of stage k+1. Therefore, to better combine with FGk of stage k global feature matrix, L2G(⋅) first repeats the tensor FLinear along the specified dimension to expand its shape. Specifically, Expand(⋅) is used to copy the tensor FLinear along the second dimension so that it matches the global feature FGk of stage k in this dimension. Then, the Concat function is used to concatenate the expanded tensor FLinear with the global feature FGk of stage k along the second dimension. The fusion operation is shown in the following Eq. (17).

(17) FGk+1=Expand(FLinear)∥FGk,k∈{1,2,3},

where ∥ represents the concatenate operation of the Concat function, Expand(⋅) represents the expansion operation.

G2L block

The G2L block transforms the input ViT feature maps into CNN feature maps. This block consists of Squeeze, Unsqueeze, Expand, Conv 1×1, and Concat. The processing of the G2L block is shown in the following Eq. (18).

(18) FLk+1=G2L(FGk),k∈{1,2,3},

where FLk+1 represents the local feature at stage k+1 and serves as the output to the G2L block, FGk represents the global feature at stage k and serves as the input of the G2L block, and G2L(⋅) represents the transformation process of the G2L block. Specifically, G2L(⋅) is a feature extractor that operates from global to local and extracts local information from the feature map of ViT. During this process, G2L(⋅) first uses the Squeeze(⋅) and Unsqueeze(⋅) functions to manipulate the dimensions of the ViT feature map.

G2L(⋅) first compresses one dimension of the tensor FGk along its second dimension using the Squeeze(⋅) function, and extracts the feature vector of the first channel, resulting in a tensor FSqueeze with a shape of 32×768, where 32 represents the number of samples and 768 represents the number of channels, as shown in the following equation (Eq. (19)).

(19) FSqueeze=Squeeze(FGk),

where, the input of the Squeeze(⋅) operation is FGk, while the output is FSqueeze.

In order to better fuse with the local features extracted by the CNN branch and obtain better feature representations, G2L(⋅) further transforms the shape of the tensor FSqueeze from 32×768 to a four-dimensional tensor with shape 32×768×8×8. Specifically, G2L(⋅) first uses two Unsqueeze(⋅) operations to add two dimensions to the FSqueeze tensor, resulting in a three-dimensional tensor with shape 32×768×1×1. Then, the Expand(⋅) operation is used to expand this tensor along the last two dimensions to a four-dimensional tensor with shape 32×768×8×8, as shown in the following equation (Eq. (20)).

(20) FExpand=Expand(Unsqueeze(Unsqueeze(FSqueeze))),

where, the input to the equation is FSqueeze and the output is FExpand, Unsqueeze(⋅) adds dimensions to the tensor, while Expand(⋅) performs an expanding operation.

In order to enhance the representational power of global features while preserving local feature information to improve the model’s performance, G2L(⋅) fuses the local features extracted from the CNN branch with global features. Specifically, G2L(⋅) concatenates the local feature FLk extracted from the CNN branch with the globally transformed feature FExpand along the channel dimension, resulting in a tensor FConcat with a shape of 32×(64+768)×8×8, where 32 represents the batch size, 64 represents the channel number of FLk, 768 represents the channel number of FExpand, and 8×8 represents the height and width of the tensor, as shown in the following equation (Eq. (21)).

(21) FConcat=FExpand∥FLk,k∈{1,2,3},

where ∥ represents the concatenate operation of the Concat function. Then, the 1×1 convolutional layer Conv2d(⋅) is applied to FConcat to perform convolution and obtain a tensor FLk+1 with a shape of 32×64×8×8, which achieves the fusion of local and global features. The final feature FLk+1 is used as the input of the CNN branch for stage k+1, as shown in the following equation (Eq. (22)).

(22) FLk+1=Conv2d(FConcat),k∈{1,2,3},

where, the input of the Conv2d(⋅) operation is FConcat, while the output is FLk+1.

Experiments

In this section, we first introduce the datasets used in this study. Then, the experiment settings are described. Finally, the results of the datasets are reported and discussed.

Datasets and settings

The study was carried out using the SEED (Zheng & Lu, 2015) and SEED-IV (Zheng et al., 2018) datasets. Both are public EEG datasets used primarily for emotion recognition. The SEED datasets contain EEG data from 15 subjects, who were presented with 15 Chinese film clips. Each clip-viewing process was divided into four stages, including a 5-s start prompt, a 4-min clip period, a 45-s self-assessment, and a 15-s rest period. EEG recordings were conducted three times on each subject, with a 2-week interval between each recording. Each recording session consisted of 15 movie clips, with each clip evoking positive, neutral, or negative emotions. The SEED-IV datasets, an extension of the SEED datasets, include 72 video clips of 2-min duration each, aimed at evoking happy, sad, fear, and neutral emotions in the subjects, who self-evaluated their emotions after watching the video clips. The EEG signals were recorded using the ESI Neuroscan system from 62 channels with a sampling rate of 1,000 Hz, which was downsampled to 200 Hz. The EEG data were filtered using a band-pass filter to remove noise and artifacts, and features such as Power Spectral Density were extracted from each segment in five frequency bands ( δ: 1∼4 Hz, θ: 4∼8 Hz, α: 8∼14 Hz, β: 14∼31 Hz, γ: 31∼50 Hz). The number of data samples in each class after the windowing operation is shown in Table 1.

Table 1 The number of data samples in each class after the windowing operation.

SEED	SEED-IV	
Negative:	50,400	Neutral:	10,170	
Neutral:	49,680	Sad:	10,245	
Positive:	52,650	Fear:	9,225	
–	–	Happy:	7,935	

We trained and tested the CIT-EmotionNet model using Tesla V100-SXM2-32 GB GPU, and implemented it using the PyTorch framework. The training was conducted using Adam optimizer, and the learning rate was set to 1e−5. The batch size was set to 32, and the dropout rate was set to 0.2. The number of classes to classify for the SEED dataset was 3, while for the SEED-IV dataset, it was 4. The samples were randomly shuffled, and the data were divided into training and testing sets with a ratio of 6:4. The cross-entropy loss function was used. The summary of hyper-parameter settings is as follows. Optimizer–Adam.

Learning rate–1e−5.

Number of classes–3 (SEED) or 4 (SEED-IV).

Batch size–32.

Loss function–cross-entropy.

Dropout rate–0.2.

Evaluation metrics

The proposed CIT-EmotionNet method will evaluate its performance based on the following metrics: average accuracy (ACC) and standard deviation (STD). The accuracy rate is defined as the ratio of correctly identified positive and negative samples to the total number of samples, as shown in Eq. (23):

(23) ACC=TP+TNTP+TN+FP+FN

where TP represents the number of predicted positive samples in the positive samples, TN represents the number of predicted negative samples in the negative samples, FP represents the number of predicted positive samples in the negative samples, and FN represents the number of predicted negative samples in the positive samples. The standard deviation is shown in Eq. (24):

(24) STD=∑i=1n(xi−x¯)2n−1

Experimental results and analysis

In order to validate the effectiveness of our proposed model, we compared the proposed model with several baseline methods on the SEED datasets and SEED IV datasets, and briefly introduced each method as follows. Support vector machine (SVM) (Zheng & Lu, 2015): A support vector machine that utilizes a linear kernel.

Dynamic graph CNN (DGCNN) (Song et al., 2018): It represented the spatial relationship between multi-channel EEG signals as a graph, and then extracted features from the graph using dynamic graph convolutional neural networks, resulting in more discriminative EEG signal features.

Bi-hemispheric discrepancy model (BiHDM) (Li et al., 2020): It used four directed recursive neural networks (RNNs) based on two spatial directions to traverse the electrode signals of two different brain regions, obtaining a deep representation of all EEG electrode signals while maintaining their inherent spatial dependencies.

Regularized graph neural network (RGNN) (Zhong, Wang & Miao, 2020): It considered the biological topological structure between different brain regions to capture local and global relationships between different EEG channels.

SST-EmotionNet (Jia et al., 2020): It integrated spatial-spectral-temporal features and used adaptive 3D attention mechanisms to explore discriminative local patterns.

3D CNN & PST (Liu et al., 2021): It introduced a 3D CNN-based method that includes a positional-spectral-temporal attention module. This module is composed of positional attention, spectral attention, and temporal attention modules, which respectively explore different brainwave features.

EEG emotion Transformer (EeT) (Liu et al., 2022): It proposed a transformer-based framework for EEG emotion recognition that exclusively utilizes self-attention blocks.

Joint-Dimension-Aware Transformer (JDAT) (Wang et al., 2023): It proposed a transformer-based model for EEG emotion recognition. By applying adaptive MSA to multidimensional features, It could focus on different EEG information, including spatial, frequency, and temporal features.

MD-AGCN (Li, Wang & Lu, 2021): It proposed a method called the Multi-Domain Adaptive Graph Convolutional Network for EEG signal analysis, which utilized the complementary knowledge between different domains of EEG signals and the topological structure of EEG channels.

4D-aNN (Xiao et al., 2022): It proposed an attention-based 4D neural network for EEG emotion recognition. It utilized CNN to process the 4D spectral-spatial representation of EEG signals. Meanwhile, it integrated the temporal attention mechanism into the bidirectional long short-term memory to explore the temporal dependency of the 4D spectral-spatial representation.

MDGCN-SRCNN (Bao et al., 2022): It combined graph convolutional networks (GCN) and convolutional neural networks (CNN). GCN learned spatial features at different levels, while CNN learned abstract features. The fully connected layer was used to fuse shallow spatial features with deep abstract features to identify highly discriminative features for emotion classification.

CLISA (Shen et al., 2023c): A contrastive learning method is proposed to enhance the generalization ability of the emotion recognition model. This method minimizes subject differences by maximizing the similarity of EEG signal representations from different subjects when exposed to the same emotional stimuli.

3D-CNN-ELM (Yuvaraj et al., 2023): It proposed using a three-dimensional convolutional neural network (3D-CNN) to recognize emotions from the spatial-temporal representation of EEG signals. First, the raw EEG signals were transformed into a 3D spatial-temporal representation to better capture spatial and temporal information between electrodes. Then, emotion recognition was conducted using 3D-CNN, leveraging its capability to learn spatial and temporal features.

EEGformer (Wan et al., 2023): It introduced a transformer-based EEG analysis model called EEGformer. EEGformer performed unified feature extraction and encoding of EEG signals by employing sequence-based transformers for synchronization, localization, and temporal transformation, thereby extracting global features of EEG signals.

Table 2 reports the average accuracy (ACC) and standard deviation (STD) of these methods and the proposed CIT-EmotionNet model for EEG emotion recognition. The SST-EmotionNet utilized a 3D CNN-based method and employed spatial-spectral-temporal features, achieving an ACC of 96.02% and 84.92% on the SEED dataset and SEED IV dataset, respectively. However, the SST-EmotionNet did not consider global features. EeT used a Transformer-based method and thus performed better than CNN-based methods on the large SEED dataset, achieving an ACC of 96.28%. However, EeT ignored local features and performed worse on the smaller SEED IV dataset, achieving an ACC of 83.27%. MDGCN-SRCNN is a combined method based on GCN and CNN that considers the fusion of different features, achieving an ACC of 95.08% and 85.22% on the SEED dataset and SEED IV dataset, respectively. In general, CIT-EmotionNet integrates local and global features well, enabling it to capture valuable features from EEG signals for emotion recognition comprehensively. Compared to the baseline model, CIT-EmotionNet further improves its accuracy. It achieved an ACC of 98.57% and an STD of 1.38% on the SEED dataset and an ACC of 92.09% and an STD of 5.33% on the SEED IV dataset, outperforming the current state-of-the-art methods.

Table 2 Performance comparison between the baseline methods and the proposed CIT-EmotionNet on the SEED and SEED-IV datasets.

Method	Year	SEED	SEED-IV	
		ACC (%) ↑	STD (%) ↓	ACC (%) ↑	STD (%) ↓	
SVM (Zheng & Lu, 2015)	2015	83.99	9.72	56.61	20.05	
DGCNN (Song et al., 2018)	2018	90.40	8.49	69.88	16.29	
BiHDM (Li et al., 2020)	2019	93.12	6.06	74.35	14.09	
RGNN (Zhong, Wang & Miao, 2020)	2020	94.24	5.95	79.37	10.54	
SST-EmotionNet (Jia et al., 2020)	2020	96.02	2.17	84.92	6.66	
3D CNN & PST (Liu et al., 2021)	2021	95.76	4.98	82.73	8.96	
JDAT (Wang et al., 2023)	2021	97.30	1.74	–	–	
MD-AGCN (Li, Wang & Lu, 2021)	2021	94.81	4.52	87.63	5.77	
EeT (Liu et al., 2022)	2022	96.28	4.39	83.27	8.37	
4D-aNN (Xiao et al., 2022)	2022	96.25	1.86	86.77	7.29	
MDGCN-SRCNN (Bao et al., 2022)	2022	95.08	6.12	85.52	11.58	
CLISA (Shen et al., 2023c)	2023	86.40	6.40	–	–	
3D-CNN-ELM (Yuvaraj et al., 2023)	2023	90.85	14.45	83.71	11.92	
EEGformer (Wan et al., 2023)	2023	91.58	2.77	–	–	
CIT-EmotionNet	2023	98.57	1.38	92.09	5.33	

Ablation experiments

In order to validate the effects of different components in our model on the EEG emotion recognition tasks, we performed ablation experiments on both the SEED and SEED IV datasets. Three ablation experiments were conducted. The first ablation experiment was conducted to the effectiveness of the fusion and interaction of local and global features. The second ablation experiment was conducted to evaluate the CIT module, aiming to validate the effectiveness of each component. The third ablation experiment investigates the impact of CIT module quantity on the performance of CIT-EmotionNet.

Ablation experiments on the major components of CIT-EmotionNet

CIT-EmotionNet consists of three main components: the CNN branch, the Transformer branch, and the CIT module. In order to validate the effectiveness of the fusion and interaction of local and global features, we conducted ablation experiments on the major components of CIT-EmotionNet. Table 3 illustrates the impact of different components of CIT-EmotionNet on EEG emotion recognition tasks. “With only CNN branch” indicates the use of only the CNN branch, which means using ResNet alone for the EEG emotion recognition task. “With only Transformer branch” indicates the use of only the Transformer branch, which means using ViT alone for the EEG emotion recognition task. “Baseline” refers to the removal of all CIT modules, and only a simple concatenation fusion is performed on the output of the feature by the CNN branch, and the Transformer branch.

Table 3 Ablation experiments on the major components of CIT-EmotionNet.

Method	SEED	SEED-IV	
	ACC (%) ↑	STD (%) ↓	ACC (%) ↑	STD (%) ↓	
With only CNN branch	74.36	8.72	50.93	18.56	
With only Transformer branch	91.77	3.98	82.87	7.18	
Baseline	93.39	3.66	87.81	6.63	
CIT-EmotionNet	98.57	1.38	92.09	5.33	

“With only CNN branch” achieves an ACC of 74.36% and an STD of 8.72% on SEED datasets, while achieving an ACC of 50.93% and an STD of 18.56% on SEED IV datasets. “With only Transformer branch” achieves an ACC of 91.77% and an STD of 3.98% on SEED datasets, while achieving an ACC of 82.87% and an STD of 7.18% on SEED IV datasets. This indicates that Transformer performs better than CNN on the EEG emotion recognition task. “Baseline” achieved an accuracy of 93.39% and a standard deviation of 3.66% on the SEED dataset, while on the SEED IV dataset, the accuracy and standard deviation were 87.81% and 6.63%, respectively. These results indicate that the combination of local and global features contributes to improving the recognition performance of the model.

We present the confusion matrices of the CNN branch, Transformer branch, Baseline, and CIT-EmotionNet models on the SEED dataset in Fig. 6. These matrices illustrate the classification performance of each model across different emotion categories. The experimental results show that CIT-EmotionNet outperforms the CNN branch, Transformer branch, and Baseline models in recognizing negative, neutral, and positive emotions. Furthermore, on the SEED dataset, CIT-EmotionNet demonstrated superior performance in recognizing positive emotions when compared to negative and neutral emotions.

Figure 6 The confusion matrices for the SEED dataset.

(A) The confusion matrix for the CNN branch and (B) the Transformer branch. (C) Confusion matrix for the Baseline model and (D) for the CIT-EmotionNet. The figures depict the classification performance of the models across different emotion categories.

Additionally, in Fig. 7, we present the confusion matrices of the CNN branch, Transformer branch, Baseline, and CIT-EmotionNet models on the SEED IV dataset, illustrating their respective classification performance across different emotion categories. The experimental results demonstrate that CIT-EmotionNet outperforms the CNN branch, Transformer branch, and Baseline models in recognizing sad, neutral, fear, and happy emotions. However, in contrast to recognizing emotions such as sad, neutral, and happy, CIT-EmotionNet exhibits poorer performance in recognizing fear emotions on the SEED IV dataset.

Figure 7 The confusion matrices for the SEED IV dataset.

(A) Confusion matrix for the CNN branch and (B) for the Transformer branch. (C) Confusion matrix for the Baseline model and (D) for the CIT-EmotionNet. These matrices depict the classification performance of the models across different emotion categories.

Ablation experiments on the major components of CIT module

To validate the effectiveness of the L2G and G2L blocks in the CIT module, ablation experiments were conducted on both blocks. Additionally, ablation experiments were performed to investigate the impact of the Transformer feature map and CNN feature map on the L2G and G2L blocks, respectively. Table 4 presents the effects of the major components of the CIT module on EEG emotion recognition tasks. “With only L2G block” refers to the integration of local features into global features, along with the fusion of Transformer Feature Maps in each stage. “With only L2G block and w/o TFM” means that only the interaction between local and global features was added on top of concatenating the output of the feature by the CNN and Transformer branches, without fusing the Transformer feature map (TFM) at each stage. “With only G2L block” refers to the integration of global features into Local features, along with the fusion of CNN Feature Maps in each stage. “With only G2L block and w/o CFM” means that only the interaction between global and local features was added on top of concatenating the output of the feature by the CNN and Transformer branches, without fusing the CNN feature map (CFM) at each stage.

Table 4 Ablation experiments on the major components of CIT module.

Method	SEED	SEED-IV	
	ACC (%) ↑	STD (%) ↓	ACC (%) ↑	STD (%) ↓	
With only L2G block	96.92	2.08	90.37	6.13	
With only L2G block and w/o TFM	78.83	7.49	58.18	11.35	
With only G2L block	97.53	1.81	91.29	5.62	
With only G2L block and w/o CFM	96.65	3.04	89.39	6.29	
CIT-EmotionNet	98.57	1.38	92.09	5.33	
Note:

The proposed module is shown in bold.

“With only L2G block”, the ACC was 96.92% and 90.37% on the SEED dataset and SEED IV dataset, respectively, indicating that using only L2G block cannot achieve satisfactory recognition performance. On the other hand, “With only L2G block and w/o TFM”, the ACC was 78.83% and 58.18% on the SEED dataset and SEED IV dataset, respectively, suggesting that incorporating TFM in L2G block can improve the accuracy of the model. “With only G2L block”, the ACC on the SEED dataset and SEED IV dataset were 97.53% and 91.29%, respectively, indicating that using only the G2L block cannot achieve satisfactory recognition performance. “With only G2L block and w/o CFM”, the ACC on the SEED dataset and SEED IV dataset were 96.65% and 89.39%, respectively, suggesting that incorporating CFM in the G2L block can improve the accuracy of the model.

Ablation experiments on the number of CIT modules

To investigate the impact of the number of CIT modules and their corresponding stages on the recognition accuracy of the model, ablation experiments were conducted with varying numbers of CIT modules at different stages on both the SEED and SEED IV datasets. The specific performance results are shown in Fig. 8. “Baseline” means no CIT module is added. “S1” represents adding CIT module only in stage 1. “S2” represents adding CIT module only in stage 2. “S3” represents adding CIT module only in stage 3. “S1 & S2” represents adding CIT module in both stage 1 and stage 2. “S1 & S3” represents adding CIT module in both stage 1 and stage 3. “S2 & S3” represents adding CIT module in both stage 2 and stage 3. It can be seen from the figure that adding CIT module in stage 3 performs better on SEED dataset, while adding CIT module in stage 2 performs better on SEED IV dataset. Meanwhile, adding CIT module in all stages, which is the strategy of Ours, performs best on both SEED datasets and SEED IV datasets.

Figure 8 Ablation experiments on the number of CIT modules.

To investigate the effect of the number of CIT modules and their corresponding stages on the recognition accuracy of the model, ablation experiments were conducted with different numbers of CIT modules and at different stages. The results indicated that the highest recognition accuracy was achieved when the CIT modules were added to stage 1, stage 2, and stage 3, suggesting that incorporating an appropriate number of CIT modules can effectively improve the performance of the model.

Conclusion

In this article, we propose a novel Convolution Interactive Transformer Network for EEG Emotion Recognition, referred to as CIT-EmotionNet, which effectively integrates the local and global features of EEG signals and uses a parallel approach of CNN and Transformer for EEG emotion recognition tasks. Firstly, we transformed the PSD features of EEG signals into spatial-spectral representations, which were used as the input of the proposed model. Then, to extract both local and global features from EEG signals simultaneously, we designed a parallel structure of CNN and Transformer, which were unified in the same framework. Finally, we developed the CIT module to facilitate the interaction and fusion of local and global features, thereby enhancing the ability of the model to extract local and global characteristics from EEG spatial-spectral representations. The CIT module was utilized in three stages of CNN and Transformer. The proposed CIT-EmotionNet achieved average recognition accuracy of 98.57% and 92.09% on the SEED and SEED-IV datasets, respectively, outperforming the state-of-the-art methods. To verify the effects of different components in CIT-EmotionNet on the EEG emotion recognition task, we conducted ablation experiments on the SEED and SEED-IV datasets. The experimental results demonstrated that the CIT module was beneficial to the fusion and interaction of local and global features, leading to an improvement in the recognition performance.

Future work. The proposed CIT-EmotionNet provides a new approach for emotion recognition based on EEG signals. This method can also be easily applied to other EEG classification tasks, such as motor imagination and sleep stage classification. In future work, we will explore the compression and acceleration of CIT-EmotionNet to improve its practicality and generalization ability.

The authors would like to express their sincere gratitude to Shanghai Jiao Tong University for the provision of the Emotion EEG Datasets. Additionally, we deeply appreciate the valuable suggestions and insights offered by the peer reviewers. We used generative AI, specifically ChatGPT, but only for grammar correction in the manuscript.

Additional Information and Declarations

Competing Interests

Author Contributions

Data Availability

The authors declare that they have no competing interests.

Wei Lu conceived and designed the experiments, performed the experiments, analyzed the data, performed the computation work, prepared figures and/or tables, authored or reviewed drafts of the article, and approved the final draft.

Lingnan Xia performed the experiments, performed the computation work, prepared figures and/or tables, and approved the final draft.

Tien Ping Tan conceived and designed the experiments, authored or reviewed drafts of the article, and approved the final draft.

Hua Ma performed the experiments, analyzed the data, performed the computation work, prepared figures and/or tables, and approved the final draft.

The following information was supplied regarding data availability:

The code is available at GitHub and Zenodo:

- https://github.com/shawnlyy/CIT-EmotionNet/tree/main

- Wei, L. (2024). CIT-EmotionNet. Zenodo. https://doi.org/10.5281/zenodo.13821897.

The SEED dataset from the BCMI Laboratory is available at Shanghai Jiao Tong University: https://bcmi.sjtu.edu.cn/home/seed/downloads.html#seed-access-anchor.

To access SEED and SEED IV, users must download and complete the license agreement on the website, and upload it to the application page. After reviewing your application, the data managers will send the download link and password via email. If you have any questions, you can contact this email: seed2022@sjtu.edu.cn.

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
