# Peer review of "CIT-EmotionNet: convolution interactive transformer network for EEG emotion recognition"

_PeerJ Computer Science, doi:10.7717/peerj-cs.2610_

## Round 0.1 · original submission · Major Revisions

The reviewer(s) have mentioned several critical comments on your article. Please carefully revise your paper based on their comments.

·

Basic reporting

Ensure grammatical accuracy and clarity in your explanations for better reader comprehension. We recommend a thorough review and revision of the manuscript to improve grammatical accuracy and enhance sentence structures. Pay careful attention to language expression for clarity. The English language of the manuscript needs revision. The grammar throughout should be consistent, for example, using either the simple present tense or the present perfect tense consistently. We recommend that the authors conduct a thorough review and revision of the manuscript to ensure consistency throughout the content. For example, the capitalization of the first letter following "where" in equations is inconsistent.

Experimental design

The manuscript lacks the necessary explanations for metrics such as ACC and STD. It needs to clearly articulate to the readers how ACC and STD are calculated. The manuscript discusses the contributions of the proposed model; however, to ensure its impact on the field, I suggest expanding the scope of model comparisons to include recent related research, such as studies from 2023. By comparing the proposed method with these recent works, the authors can better illustrate the novelty and improvements their model offers.

Validity of the findings

While the authors have extensively discussed the novelty of their proposed model, I recommend a more explicit description of the unique contributions of the proposed model and its specific advancements over existing methods. Emphasizing these contributions will enhance the manuscript's novelty and clarify its significance to the field.

Additional comments

The manuscript lacks explanation regarding the use of pre-trained models. It does not clarify whether the ViT model and the ResNet model utilize pre-trained versions. Please provide a comprehensive description of these aspects to enhance the reproducibility and validity of the experiments in the manuscript. The manuscript provides information on the process of extracting EEG local features and EEG global features by the CIT-EmotionNet model. However, I suggest further elaboration on what exactly the EEG local features and EEG global features refer to in terms of aspects of the EEG signals.

Reviewer 2 ·

Basic reporting

The paper is generally well-organized and written, with clear and unambiguous professional English used throughout. Sufficient references were considered, providing a solid field background and context. The findings of the experiments were effectively compared with these references.

The figures and tables are presented in a clear and understandable manner, aiding in the comprehension of the study's results.

The results and hypotheses are well-understood, and the paper is self-contained with relevant results aligned with the hypotheses.

However, the method and experiments section includes excessive detail, which may benefit from clarification to streamline the content and improve readability.

Overall, the paper demonstrates a professional article structure, with raw data shared for transparency and reproducibility.

Experimental design

The experiments were conducted to find the solution for the research question. The methods were described in detail.

A few comments are given as follows:
* In Section 0.4, the representation of feature vector is given briefly. It is explained in Section 0.5 later. It seems confusing. Instead, you can explain verbally in 0.4 and explain completely in 0.5.
* PSD formulation may be given to be self-contained.
* The feature calculation needs to be clearly summarized. It would be better to use a graphical representation.
* Superscript M is not explained in section 0.5.
*At each stage, how and why these parameters selected should be briefly explained.
* Xmain, Xshortcut etc. can be denoted in Figure for better understanding.
* The PSD generated matrix of the same 4 sec sample is used for both local and global feature extraction. Some abstract 32x32x5 form of features is supplied to Transformer block. Is it this compressed really reflecting the global mapping of brain regions?
* The representation of CIT algorithm seems too complex.
*The differential entropy is not mentioned until Experiments section. Is it used in feature extraction stage?

Validity of the findings

The paper includes detailed tables and graphs. The findings are compared with the state-of-the-art models.
A few remarks includes the followings:
* The the number of data in each class after windowing operation should be given in a Table.
*I would expect to see your results before the comparison table. May be you can first give the some confusion matrices, results of your complete algorithm and the ablation experiments. Then you can compare your results in a Discussion section.
* The accuracy alone is not sufficient for performance measure. You should include Precision, Recall and F1 score also.
*It might be better to add a paragraph on the computational complexity or the execution times.

Reviewer 3 ·

Basic reporting

The paper utilizes clear, unambiguous, and professional English throughout. The language is precise and technical, appropriate for the target audience of researchers and professionals in the field of affective computing and machine learning.The introduction effectively establishes the context of the research by discussing the significance of emotion recognition in affective computing. It highlights the challenges in using EEG signals for emotion recognition and the need for integrating global and local features to improve performance. The literature review is comprehensive and well-referenced, including relevant studies on CNN and Transformer models in EEG emotion recognition. However, I suggest comparing the proposed model with recently published work in the filed of EEG-based affective computing:

1. Shen, X., Liu, X., Hu, X., Zhang, D. and Song, S., 2022. Contrastive learning of subject-invariant EEG representations for cross-subject emotion recognition. IEEE Transactions on Affective Computing.
2. Gilakjani, S.S. and Al Osman, H., 2022, August. Emotion Classification from Electroencephalogram Signals Using a Cascade of Convolutional and Block-Based Residual Recurrent Neural Networks. In 2022 IEEE Sensors Applications Symposium (SAS) (pp. 1-6). IEEE.
3. Gilakjani, S.S. and Al Osman, H., 2023. A Graph Neural Network for EEG-Based Emotion Recognition with Contrastive Learning and Generative Adversarial Neural Network Data Augmentation. IEEE Access.

The figures in the paper are relevant, high-quality, and well-labeled.

Experimental design

The experimental design of the paper is well-documented, and thorough. It covers all necessary aspects to evaluate the proposed CIT-EmotionNet model effectively. However, there are some minor issues that should be addressed. First, I suggest comparing your results with the papers I recommended. This would provide a broader context for evaluating the performance of your model and demonstrate its competitiveness against a wider range of existing methods. Second, I recommend separating a portion of the dataset (e.g., 20%) as a testing dataset that remains unseen during the training phase. This is relatively a robust way to ensure that your model generalizes well to unseen data. Having only a training dataset or performing cross-validation alone will not sufficiently demonstrate that your model outperforms others in terms of generalization to new, unseen data.

Validity of the findings

The paper presents a novel approach by integrating Convolutional Neural Networks (CNN) and Transformer models for EEG emotion recognition. This combination leverages both local and global features, which is an innovative aspect that addresses current challenges in the field. However, the impact and novelty of the findings could be further emphasized by discussing how this approach advances the state-of-the-art and its potential applications in real-world scenarios. Including comparisons with additional recent and impactful studies would strengthen this assessment. All underlying data used in the study are derived from publicly available datasets (SEED and SEED-IV), and the preprocessing steps are thoroughly described.
Overall, the findings of the paper are valid, robust, and statistically sound. The novelty of integrating CNN and Transformer models for EEG emotion recognition is a significant contribution to the field.

---

## Round 0.2 · Major Revisions

The authors are requested to further carefully address the reviewer comments to improve the quality of the paper.

·

Basic reporting

The authors have resolved all of my previous issues. In this study, the authors propose a novel Convolution Interactive Transformer Network for EEG Emotion Recognition, which eûciently integrates the global and local features of EEG signals. Initially, the authors convert the raw EEG signals into spatial-spectral representations, which serve as the inputs into the model. The model integrates Convolutional Neural Network (CNN) and Transformer within a single framework in a parallel manner. the authors propose a Convolution Interactive Transformer module, which facilitates the interaction and fusion of local and global features extracted by CNN and Transformer respectively, thereby improving the average accuracy of emotion recognition.

Authors should discuss some recent papers on local and global EEG in the introduction or related work, such as [1-2].
[1] Learning Robust Global-Local Representation from EEG for Neural Epilepsy Detection
[2] Emotion recognition using hierarchical spatial–temporal learning transformer from regional to global brain

Experimental design

The manuscript has a certain degree of innovation. The research question has a clear definition. The method description has sufficient details.

Validity of the findings

N/A

Additional comments

N/A

Reviewer 2 ·

Basic reporting

The paper describes the methodology and the results in a clear way. Sufficient references were covered.
A few minor comments as follows:
*In lines 77-82 local and global features are mentioned repeatedly. You may clarify the sentences.
* The contributions starting from line 88 and 92 seems so similar.
* It would be better to reconsider the explanations to give the message clearly, since too much details are included with some repetitions in the paper.

Experimental design

The original contribution of the paper is explained in detail. The research area fall into the scope of the journal. The standards of a research paper is satisfied with the completed revisions.

Validity of the findings

The data is carefully collected and the results were presented in detail. The algorithm and the findings seem robust.

Reviewer 3 ·

Basic reporting

The paper utilizes clear, unambiguous, and professional English throughout. The language is precise and technical, appropriate for the target audience of researchers and professionals in the field of affective computing and machine learning.

Experimental design

The experimental design of the paper is well-documented and thorough, covering all necessary aspects to effectively evaluate the proposed CIT-EmotionNet model. However, there are still some issues that need to be addressed. I previously suggested reserving a portion of the data to evaluate the model's generalizability, but this has not been implemented. The training and testing split for each subject resembles cross-validation, which does not clearly demonstrate the true performance of the proposed model. Additionally, although I recommended comparing the results with three recent papers, they have only done so with one. I would like to see how their model performs against a Transformer-based emotion recognition model and a contrastive learning-based model in the same field.

Validity of the findings

The paper introduces a novel approach by integrating Convolutional Neural Networks (CNN) and Transformer models for EEG emotion recognition, effectively leveraging both local and global features. This innovative combination addresses current challenges in the field. However, the impact and novelty of the findings could be further highlighted by discussing how this approach advances the state-of-the-art and its potential real-world applications. Including comparisons with additional recent and influential studies would further strengthen the assessment.

Overall, the findings are valid, robust, and statistically sound. The integration of CNN and Transformer models for EEG emotion recognition represents a significant contribution to the field.

---

## Round 0.3 · Minor Revisions

Please address the reviewers' final comments carefully.

Reviewer 3 ·

Basic reporting

The paper is written in clear, unambiguous, and professional English, with precise and technical language suitable for its target audience of researchers and professionals in affective computing and machine learning. The introduction effectively sets the research context by discussing the importance of emotion recognition in affective computing, emphasizing the challenges of using EEG signals for this purpose, and underscoring the need to integrate global and local features to enhance performance. The literature review is comprehensive, well-cited, and includes relevant studies on the application of CNN and Transformer models in EEG-based emotion recognition. The paper also incorporates recommendations for comparison and related work.
There is only one typo in line 126 of the trackchanges file (0differential) that needs to be corrected.

Experimental design

The experimental design in the paper is well-documented and thorough, covering all essential aspects for effectively evaluating the proposed CIT-EmotionNet model. I appreciate that you have considered two testing isolation scenarios to demonstrate the generalizability of your model. While implementing previous models in this format and testing them all in an identical setting can be time-intensive, I recommend adding a separate section dedicated exclusively to showcasing the results of your proposed model in this setup. This would provide a clear perspective on how your model generalizes to unseen datasets, emphasizing its robustness and applicability.

Validity of the findings

The paper introduces an innovative approach by combining Convolutional Neural Networks (CNN) and Transformer models for EEG-based emotion recognition. This integration effectively captures both local and global features, addressing key challenges in the field. Overall, the findings are valid, robust, and statistically sound. The novel integration of CNN and Transformer models for EEG emotion recognition represents a meaningful advancement in the field.

---

## Round 0.4 · accepted · Accept

Thank you for addressing all the comments from the reviewer. I carefully reviewed the author's responses and believe they answered well.